# Radiation Dermatitis: Radiation-Induced Effects on the Structural and Immunological Barrier Function of the Epidermis

**DOI:** 10.3390/ijms25063320

**Published:** 2024-03-15

**Authors:** Claudia E. Rübe, Benjamin M. Freyter, Gargi Tewary, Klaus Roemer, Markus Hecht, Christian Rübe

**Affiliations:** 1Department of Radiation Oncology, Saarland University Medical Center, 66421 Homburg, Saar, Germanygargi.tewary@uks.eu (G.T.);; 2José Carreras Center, Internal Medicine, Saarland University Medical Center, 66421 Homburg, Saar, Germany

**Keywords:** radiation dermatitis, ionizing radiation, skin inflammation, epidermis, cellular senescence, senescence-associated secretory phenotype (SASP)

## Abstract

An important hallmark of radiation dermatitis is the impairment of the mitotic ability of the stem/progenitor cells in the basal cell layers due to radiation-induced DNA damage, leading to suppressed cell renewal in the epidermis. However, this mechanism alone does not adequately explain the complex pathogenesis of radiation-induced skin injury. In this review, we summarize the latest findings on the complex pathogenesis of radiation dermatitis and correlate these with the clinical features of radiation-induced skin reactions. The current studies show that skin exposure to ionizing radiation induces cellular senescence in the epidermal keratinocytes. As part of their epithelial stress response, these senescent keratinocytes secrete pro-inflammatory mediators, thereby triggering skin inflammation. Keratinocyte-derived cytokines and chemokines modulate intercellular communication with the immune cells, activating skin-resident and recruiting skin-infiltrating immune cells within the epidermis and dermis, thereby orchestrating the inflammatory response to radiation-induced tissue damage. The increased expression of specific chemoattractant chemokines leads to increased recruitment of neutrophils into the irradiated skin, where they release cytotoxic granules that are responsible for the exacerbation of an inflammatory state. Moreover, the importance of IL-17-expressing γδ-T cells to the radiation-induced hyperproliferation of keratinocytes was demonstrated, leading to reactive hyperplasia of the epidermis. Radiation-induced, reactive hyperproliferation of the keratinocytes disturbs the fine-tuned keratinization and cornification processes, leading to structural dysfunction of the epidermal barrier. In summary, in response to ionizing radiation, epidermal keratinocytes have important structural and immunoregulatory barrier functions in the skin, coordinating interacting immune responses to eliminate radiation-induced damage and to initiate the healing process.

## 1. Clinical Background

Radiation dermatitis is a common side effect of radiotherapy (RT) in cancer patients [1]. Despite substantial technological advancements in the application of ionizing radiation (IR), radiation-induced skin damage during RT of certain tumor entities, such as the head and neck, as well as breast cancer, remains a significant issue [2,3]. Precision radiotherapy has improved the therapeutic efficacy of cancer treatment, either alone or in combination with other treatment modalities, such as immunomodulatory drugs. Immune checkpoint inhibitors have revolutionized cancer treatment in recent years and offer new opportunities to treat a variety of tumors [4], but they also pose a risk of dermatological side effects [5]. Currently, the impact of combining immune checkpoint inhibitors with radiotherapy on normal or tumor tissue reactions is not fully understood, especially whether it may lead to an increased incidence of dermatologic adverse events [6]. However, to assess the impact of these immunomodulatory agents on radiation-induced skin and tumor reactions, it is necessary to understand the complex pathogenesis of radiation dermatitis at the cellular and molecular levels [7]. In human skin, the epidermis forms a multi-layered tissue through the differentiation of keratinocytes and thereby provides interconnected mechanical and immunological protection. Epidermal and follicular keratinocytes interact with multiple immune cells to shape, amplify and regulate inflammatory responses, thus creating the first line of defense against environmental threats [7]. Over the past few years, technical advances in molecular biology and the use of innovative experimental models have led to a better understanding of the pathogenesis of numerous dermatological diseases, including cutaneous radiation injuries [8,9,10]. In this review, we describe the pathomechanisms of radiation-induced skin damage based on the latest findings, with a particular focus on the structural and immunological barrier function of the epidermis.

## 2. Anatomy and Physiology of Human Skin

The human skin, as the largest organ in the human body, covers a surface area of ~1.8 m^2^ and provides essential protective functions against external physical, chemical and biological threats (Figure 1). In addition to its endocrine and exocrine activities, the skin plays a crucial role in thermoregulation and sensory perception [11]. The human skin consists of the epidermis; the external, stratified, non-vascularized epithelium and the underlying dermis, composed of mesh-like collagenous connective tissue rich in blood vessels and nerves [12]. The epidermis, the outermost layer of the skin, forms a waterproof, protective covering over the body surface and consists of a stratified squamous epithelium with an underlying basal lamina (Figure 2). Depending on the anatomic location, the epidermis can range from 0.05 to 1.5 mm in thickness (eyelid–palms and soles), with keratinocytes being the predominant cell type, accounting for over 90% of the cells. Epidermal stem cells in the stratum basale give rise to progenitor cells that undergo a serious of differentiation processes. As keratinocytes mature, they migrate towards the skin surface, flatten out and eventually shed in the horny layer. These epigenetically regulated differentiation processes characterize the tissue architecture of the epidermis, forming the suprabasal stratum spinosum, stratum granulosum and terminally differentiated stratum corneum [13]. Throughout this programmed transformation, differentiated keratinocytes undergo nuclear degradation and cell flattening in the stratum granulosum. The stratum corneum is composed of keratinized corneocytes lacking organelles, connected by corneodesmosomes and eventually shed from the skin surface [14]. While corneocytes are continuously shed from the horny layer during their upward maturation, the deeper layers remain tightly connected by corneodesmosomes and intercellular lipids [15]. Maintaining the integrity of the stratum corneum is crucial to reducing transepidermal water loss. Sebaceous glands in the dermis produce sebum, an emulsion of water and oil, which reaches the surface via hair follicles, forming a hydrolipidic film [16]. This film, along with the acidification of the horny layer, is essential to the skin’s physiological microbiota and barrier function [17]. Since the epidermis lacks blood vessels, the nutrient supply for epidermal homeostasis relies on diffusion from the dermis.

The human dermis (3–5 mm thick) is closely attached to the epidermis through the basement membrane and contains numerous nerve endings responsible for sensations of heat and touch. Structurally, the human dermis is divided into the papillary and reticular layers. The papillary layer is made up of loose areolar connective tissue with finger-like projections called papillae that strengthen the connection between the two skin layers. The reticular layer consists of dense connective tissue with collagen, elastic and reticular fibers that ensure the strength, extensibility and elasticity of the skin. This layer also houses hair follicles, sebaceous glands, sweat and apocrine glands and lymphatic and blood vessels. The well-capillarized vessel system in the dermis plays a crucial role in nourishing the epidermis [18] and in the development of inflammatory skin conditions [19].

It is important to acknowledge that dermatological studies are mainly conducted on the back skin of mice. Unlike in murine skin, the interfollicular epidermis makes up a greater proportion of human skin, with only sparse hair follicles [20]. Additionally, human skin has a thicker dermis and a multilayered epidermis compared to murine skin. These anatomical and physiological differences should be taken into consideration when interpreting the results from mouse models to gain a better understanding of the underlying mechanisms of human skin diseases [21].

## 3. Clinical Features of Radiation-Induced Skin Reactions

The clinical manifestations of radiation-induced skin reactions can be categorized into acute reactions, which occur within 90 days of starting treatment, and late effects, which appears months to years after completing RT [22]. Radiation dermatitis is more common in patients undergoing RT for head and neck cancer, breast cancer or skin cancer due to the higher dose of radiation to the skin. Factors such as the radiation quality, total dose, dose rate and dose per fraction, volume and surface area exposed to IR and the use of bolus material can influence the risk of developing radiation dermatitis [22].

Certain skin areas, such as the front of the neck, chest, abdomen and flexor sides of the extremities, are more sensitive to radiation. In breast cancer patients, larger breast size, breast reconstruction and implants are associated with an increased risk of more severe radiation dermatitis [23]. Due to tissue overlay, the axillae and sub-mammary regions are more prone to friction and moist reactions due to the formation of a “moist chamber”. Lifestyle factors like obesity, chronic sun exposure and smoking appear to increase the risk of radiation dermatitis [22]. In addition, the skin microbiome, particularly its colonization by *Staphylococcus aureus*, may play a role in the development of severe radiation dermatitis [24]. Patients with genetic disorders affecting their DNA repair capacity are at a higher risk of developing severe radiation dermatitis [22,25]. Patients receiving concurrent chemotherapy [26] or targeted cancer therapy [27] are also more susceptible to pronounced radiation dermatitis.

Acute radiation dermatitis usually develops gradually during conventionally fractionated radiotherapy, with the timing of onset varying from days to weeks after the start of radiation treatment. The changes in the skin depend not only on the IR parameters but also on the individual’s skin sensitivity [28] and can include erythema, edema, changes in pigmentation, hair loss and dry or moist desquamation [22]. In external beam RT with kilovoltage X-rays, erythema occurs at skin doses ≥ 6 Gy, dry desquamation at skin doses ≥ 20 Gy and moist desquamation at skin doses ≥ 30 Gy [22]. Radiation of higher energies requires higher doses to produce the same level of skin damage, as the maximum dose is received in deeper tissues below the skin. Acute dermatitis typically progresses for 7 to 10 days after RT ends, after which re-epithelialization begins unless a bacterial infection is present. Small areas of the skin can tolerate higher doses than large regions, as the epidermis can heal due to keratinocytes migrating from the surrounding healthy skin during the recovery phase [29]. Severe dermatitis with ulceration of the epidermis or underlying dermis is linked to prolonged inflammation and a prolonged healing time and may lead to significant scarring [22].

The severity of radiation dermatitis can be evaluated using the grading system developed by the Radiation Therapy Oncology Group (RTOG) and the European Organisation for Research and Treatment of Cancer (EORTC). This system categorizes radiation dermatitis into four grades: grade 1, characterized by faint erythema with dry desquamation; grade 2, with moderate to brisk erythema and patchy, moist desquamation mostly in the skin folds; grade 3: showing confluent, moist desquamation in areas other than the skin folds; grade 4, presenting with skin ulceration with trauma-associated or spontaneous bleeding [30]. IR exposure weakens the skin’s natural defenses against microbes, increasing the risk of bacterial infections, commonly caused by *S. aureus* [24]. The itching, discomfort and pain associated with skin reactions from radiation dermatitis can significantly impact patients’ quality of life. Additionally, potential long-term damage such as cutaneous atrophy, subcutaneous fibrosis and telangiectasia can have negative cosmetic effects. Chronic skin reactions following IR exposure are based in particular on fibrosis of the cutis and subcutis. These complex molecular mechanisms of skin fibrosis are based on the excessive production of collagen and extracellular matrix production by the dermal fibroblasts. Since a comprehensive review of chronic radiation reactions would go beyond the scope of this work, we refer to the various excellent reviews on this topic.

## 4. Pathophysiology of Radiation-Induced Skin Reactions

The high-energy radiation delivered during radiotherapy causes direct and indirect ionization events that result in damage to the cellular macromolecules, primarily in the form of radiation-induced DNA damage. Through this DNA-damaging mechanism, IR exposure affects virtually every cellular component of the skin, particularly the epidermal keratinocytes, including their stem and progenitor cells [31].

### 4.1. Suppressed Proliferation of Basal Cells following IR Exposure

Radiation-induced DNA damage affects the ability of the stem/progenitor cells in the basal cell layer to divide, leading to a decrease in the renewal and survival of actively proliferating keratinocytes in the epidermis [32]. The rate of cell proliferation in the epidermis is typically measured using the labelling index in the basal layer, which indicates the proportion of cells undergoing replication. Experimental studies have shown that progenitor cells make up ≈10% of the total cell population with ≈15% cycling cells, while stem cells account for less than ≈1% of keratinocytes with ≈2% cycling cells [33,34]. These findings suggest that epidermal stem cells have a slower rate of division compared to their daughter basal cells, with an average of ≈3 transit cell divisions in the basal layer before they migrate to the upper layers of the epidermis.

The extent of cell proliferation is temporarily inhibited depending on the radiation dose administered, causing particularly radiation-sensitive proliferating cells to die through apoptosis. A few days after radiation-induced tissue damage, the increased loss of stem/progenitor cells leads to increased proliferation of the basal cells to restore skin tissue homeostasis. Following a single dose of 20 Gy, reactive hyperplasia of the epidermis with increased proliferation of keratinocytes is observed within 1–2 weeks following IR exposure [35].

### 4.2. Radiation-Induced Senescence in Keratinocytes

Senescence is a complex cellular stress response that halts cells’ ability to proliferate despite signals promoting cell division [32,33]. Senescent cells undergo dramatic changes in their chromatin organization, gene expression and metabolic activity [33,34]. While high epidermal turnover rates lead to only small accumulations of senescent keratinocytes in the human skin under normal conditions, stressors such as IR exposure can trigger premature senescence in the epidermal keratinocytes [35,36]. Various biomarkers, such as cyclin-dependent kinase inhibitors p16^INK4a^ and p21^WAF1^, as well as the expression of lysosomal β-galactosidase activity (SA-β-Gal), are often used to identify cells displaying senescence-related characteristics in skin tissues. Preclinical studies have shown increased SA-β-Gal expression in the epidermal keratinocytes following high-dose radiation (1 × 20 Gy) [37] and moderate doses (5 × 2 Gy) [38]. The biomarkers of senescence involve the upregulation of cell cycle inhibitors, the downregulation of proliferation-associated proteins and a loss of lamin B1, a key component of the nuclear lamina [39]. Analysis of irradiated skin using immunofluorescence and electron microscopy has revealed a decrease in the lamin B1 levels in the epidermal keratinocytes in response to senescence-inducing IR doses [37]. In addition to acquiring stable growth arrest, senescent cells exhibit a hypersecretory phenotype, known as senescence-associated secretory phenotype (SASP), releasing a variety of cytokines, chemokines, growth factors and proteases [35] (Figure 3).

### 4.3. Secretion of Pro-Inflammatory Mediators by Senescent Keratinocytes

Previous research has demonstrated that IR exposure not only impacts the proliferation and differentiation of keratinocytes but also influences the communication between the epidermal keratinocytes and immune cells by producing pro-inflammatory mediators as part of radiation-induced senescence [40]. Keratinocytes possess various pattern recognition receptors that can identify pathogen- and damage-associated molecular patterns associated with pathogen invasion and other stressors. This leads to the activation of second messengers through the NF-κB or MAPK pathways, the transcription of pro-inflammatory genes, cytokine signaling and cell-mediated inflammatory responses [37,41]. Pro-inflammatory cytokines and chemokines play a crucial role in activating the immune cells and significantly contribute to the development of radiation dermatitis syndrome [42]. The experimental evidence suggests that the expression of senescence-associated cytokines following IR exposure plays a key role in the complex pathogenesis of radiation dermatitis by modulating various senescence-related processes that have a major impact on skin tissue homeostasis. Within 1 week after high-dose IR to murine skin, an increased expression of multiple SASP factors can be observed through transcriptome analyses, just before noticeable skin reactions become clinically apparent [37]. Transcriptome data comparing non-irradiated with irradiated skin showed strong upregulation of multiple SASP-associated cytokines, chemokines, adhesion molecules and matrix metalloproteinases and their inhibitors following IR exposure [37]. Cell adhesion molecules and integrins play a critical role in facilitating transendothelial migration of the immune cells from the bloodstream to irradiated skin [42,43]. Furthermore, the gene expression related to inflammatory and immunoregulatory interactions, including neutrophil degranulation, is markedly increased after exposure to IR [37]. On the other hand, the gene expression related to collagen and extracellular matrix organization, keratinization and the formation of the cornified envelope is significantly decreased following IR exposure [37].

### 4.4. Immune Reactions of the Skin following IR Exposure

Unstressed skin contains a network of innate immune cells that reside in non-inflamed skin but readily respond to infected, stressed or damaged cells by secreting cytokines, thereby activating and/or recruiting both innate and adaptive immune cell populations [8]. Resident cells such as epidermal Langerhans cells, dermal dendritic cells and resident T cells are capable of initiating signaling pathway responses to barrier injury. The Langerhans cells located in the stratum spinosum extend their dendrites to capture antigens near the skin surface and thus serve as the immune sentries of the epidermis. When confronted with pathogens, Langerhans cells migrate from the epidermis to the lymph nodes for antigen presentation and promote both effector and regulatory T cell responses in the skin [44]. Effective skin immune responses require the formation of antigen-specific effector T cells (CD4+, CD8+, Treg), which home in on cutaneous sites of injury and actively respond to a stimulus. Dendritic cell-mediated antigen presentation leads to the activation and clonal expansion of antigen-specific CD4+ and CD8+ T cells within the skin-draining lymph nodes [45]. During priming, T cells are imprinted by the dendritic cells to express the appropriate homing molecules, including chemokine receptors, enabling effector T cell migration into the skin. Long-lasting immunity against future immune challenges is mediated by memory T cells. Following pathogen clearance, fractions of antigen-experienced T cells turn into long-lived memory T cells. The memory T cells found in the skin include both recirculating cells, which retain the potential to recirculate between the lymph, blood and non-lymphoid organs, and tissue-resident memory T cells, which remain in the skin for long periods of time and provide long-lasting protective immunity [46]. The dermis contains many unconventional lymphocyte subsets, including the αβ and γδ T cell populations, which constitute only a low percentage of all the T lymphocytes in the peripheral blood and lymphoid tissue but which are the most abundant T lymphocyte subsets in epithelial barriers such as skin [47]. These αβ and γδ T cell populations, activated by different molecular mechanisms, are both classified as part of innate immunity due to their non-MHC-restricted antigen recognition and their rapid response to invading pathogens. Increased proliferation and activation of the αβ and γδ T cells in the dermis are common features of acute and chronic skin inflammation [47]. The macrophage populations in the skin comprise both resident macrophages that recognize and respond to antigens early and circulating monocyte-derived subsets that also play an important role in transporting antigens to the draining lymph nodes [48]. Recent experimental work has shown that radiation dermatitis is characterized by the increased infiltration of Langerhans and dendritic cells (CD11+, CD25+), multiple T cell populations (CD4+, CD8+, αβ TCR+, γδ TCR+), macrophages and monocyte-derived macrophages (F4/80+; Ly6C+) [37]. Accordingly, radiation-induced skin inflammation is associated with significant changes in both the skin-resident and skin-infiltrating immune cells (Figure 3). This includes expansion of the existing immune cell populations in the skin, as well as the increased recruitment of new populations, such as neutrophils, from the blood circulation. Experimental studies have shown that radiation-induced senescence in keratinocytes is associated with the secretion of multiple chemokines, responsible for the chemotactic recruitment of inflammatory and immune cells within the dermal compartment and crucial to orchestrating complex immune responses to environmental stressors. Accordingly, it was shown that increased mRNA expression of chemokine C-X-C motif ligand 2 (Cxcl2) and 5 (Cxcl5) in irradiated skin leads to increased recruitment of neutrophils in the already inflamed skin [37]. Mature neutrophils capture invading microorganisms and destroy them through phagocytosis, with subsequent intracellular degradation [49]. Upon the detection of pathogens, neutrophiles release cytotoxic granules and form extracellular neutrophil traps [50]. The recruited neutrophils in the irradiated skin exert strong pro-inflammatory effects through the direct release of toxic effectors, such as myeloperoxidase, that are responsible for exacerbating the inflammatory state in the dermal compartments [37].

### 4.5. Structural Barrier Dysfunction of the Epidermis

During normal skin homeostasis and tissue renewal, the epigenetic mechanisms that regulate the gene expression profile of each individual cell determine the decision between the self-renewal of the epidermal stem cells and their differentiation into fully differentiated keratinocytes [14]. Specific transcription factors, governing epithelial lineage specification, stratification and barrier formation, tightly collaborate to ensure proper epidermal homeostasis [51]. The transcription factor JunB maintains the structural integrity of the epidermis following transient stress by concomitantly fine-tuning proliferation and differentiation. Radiation-induced DNA damage stress disrupts normal proliferation, differentiation and the cornification of keratinocytes within the epidermis, thereby affecting the skin’s protective barrier function. This radiation-induced injury to the epidermal integrity is associated with the dysregulation of the transcription factor JunB [37]. Following cutaneous challenge due to IR exposure, the skin tries to compensate for the cell loss by increasing the mitotic activity of the basal cells. This leads to reactive hyperplasia of the epidermis but with impaired differentiation and keratinization processes [37]. Previous studies have demonstrated the essential role of the IL-17-expressing γδ T cells in mediating the radiation-induced hyperproliferation of keratinocytes [52]. IL-17 and the transcriptional regulator IκBζ are of particular importance to the pathogenesis of psoriasis, an autoinflammatory skin disease characterized by cytokine-driven hyperproliferation of the keratinocytes [53]. In the epidermal keratinocytes, IL-17A triggers the NF-κB- and STAT3-dependent transcriptional upregulation of IκBζ expression, thereby inducing epidermal hyperproliferation. The current studies on radiation dermatitis suggest that, similar to psoriasis, there is a hyperactivation of these proinflammatory signaling pathways in irradiated skin, which leads to hyperproliferation of the keratinocytes and thus to epidermal hyperplasia [37]. Moreover, proper barrier function requires a choreographed network of cell–cell adhesion systems and their associated cytoskeletons [54]. As the keratinocytes progress towards the upper epidermis, they undergo the cornification process, involving crosslinking of the keratinocyte proteins and the breakdown of their nuclei and other organelles by intracellular and secreted proteases. Recent studies have shown that IR exposure disrupts the permeability barrier of the skin by altering the gene expression of numerous structural and junctional components required for epidermal barrier development [37]. Accordingly, following high-dose IR exposure, these fine-tuned keratinization and cornification processes are disturbed, leading to structural dysfunction of the epidermal barrier.

## 5. Radiation-Induced Alopecia

### 5.1. Clinical Features

Hair follicles (HFs) are highly sensitive to ionizing radiation, and radiation-induced alopecia is a frequent adverse effect of radiotherapy (Figure 4). Since hair is a crucial and integral part of our appearance, radiation-induced hair loss (transient, persistent, progressive alopecia) can have significant negative psychological effects on patients. The clinical presentations of radiation-induced alopecia can vary greatly, depending on the quality of the radiation beams, cumulative doses, dose fractionation and concomitant treatments [55,56]. Transient alopecia develops after single doses of 2–5 Gy, and when HFs are only mildly damaged, hair can regrow within few weeks after the last radiation session [57]. Persistent alopecia results from high-dose radiotherapy with partial hair reduction at single doses of 5–10 Gy, whereas complete hair loss can occur beyond 10 Gy [55,57]. In patients with transient alopecia, graying can also develop in the regrowing hair, which indicates profound damage to the HFs’ pigmentary units. Hair loss alone or in combination with acute radiation dermatitis begins 2–3 weeks following fractionated radiotherapy. Incomplete hair regrowth at 6 months following the completion of radiotherapy is defined as persistent alopecia, indicating that radiation has caused irreversible damage to the HFs, including their epithelial and melanocyte stem cells [55,57].

### 5.2. Anatomy and Cyclic Remodeling of the Hair Follicles

HFs represent a complex neuroectodermal–mesodermal interacting system that cyclically regresses and regenerates throughout its lifespan, undergoing transformations from active growth (anagen) to apoptosis-driven involution (catagen) and quiescence (telogen) (Figure 4). Normal hair shedding occurs at the end of the telogen phase, when a new anagen HF is already initiated. Cyclic anagen development is driven by the hair follicle stem cells (HFSC) in the bulge region [58]. The mesenchymal cells in the dermal papilla activate the epithelial HFSCs during the telogen-to-anagen transition, control hair bulb matrix cell differentiation in the anagen HFs and participate in catagen induction by interrupting morphogen secretion. During anagen development, some dermal sheath fibroblasts migrate into the dermal papilla to increase their volume and morphogenic potential. The epithelial HFSCs located in the bulge region proliferate during the early anagen phase and generate progeny, which build the outer and inner root sheaths of the anagen HF epithelium. In the full anagen phase, the HFSCs in the bulge remain proliferatively inactive, whereas the proximal hair bulb harbors maximally dividing hair matrix keratinocytes (transit-amplifying cells) that terminally differentiate to form the hair shaft. During fractionated radiotherapy, the HF cycle is negatively influenced not only by the dysfunctions of various stem cell populations but also by the signals secreted by the extrafollicular cells in the skin [58,59]. However, how the hair cycle clock is controlled at the molecular level and how IR affects HF cycling are still largely unknown [60,61]. Human scalp and mouse pelage HFs have similar but not identical anatomical structures. Most human scalp HFs (≈90%) are in the terminal anagen phase, with a length of 4–5 mm [62,63,64], whereas mouse pelage anagen HFs are much shorter (1–2 mm) [65]. The anagen and catagen phases of the HFs in the dorsal pelage of mice persist for about 18 days and 3–4 days, respectively [66]. Accordingly, murine hair cycle phases are clearly shorter, and their HFs cycle much faster than human scalp HFs [62,63]. In addition to these morphological–functional differences, human and murine HFs are subject to both shared and distinct growth controls; therefore, the research results from animal studies cannot be directly extrapolated to human responses.

### 5.3. Pathobiology of HF Cycling in Response to IR

Both epidermal and follicular keratinocytes form a structural and immunological barrier that responds to various injuries, such as IR exposure. Radiation induces DNA damage in all the cellular components of the affected skin, which subsequently triggers a series of DNA damage reactions. Radiation-induced DNA damage typically leads to p53-dependent cell cycle arrest, apoptosis or senescence, depending on the extent of DNA damage and repair efficiency and the cell cycle status [67]. The radiation sensitivity varies in the different HF compartments. In particular, various stem cell populations differ in their susceptibility to IR, depending on their cell cycle status and intrinsic regulatory networks [68]. Highly proliferating hair matrix keratinocytes are extremely radiosensitive and undergo radiation-induced apoptosis, whereas quiescent bulge HFSCs and HF melanocyte stem cells are less sensitive to radiation [65]. Quiescent bulge HFSCs express higher amounts of the anti-apoptotic protein BCL2 and show attenuated p53 activation, making them more resistant to radiation-induced apoptosis [65,69]. However, they may accumulate mutations and epigenetic alterations, which can compromise their functions [20]. Previous mouse studies with fractionated low-dose radiation have shown that CD34+ HFSCs differentiate and leave their stem cell niche with an increasing accumulation of radiation-induced DNA damage [20]. As part of this differentiation process, epigenetic characteristics and cell identities are changed to meet the new functional requirements [34]. 

In anagen HFs, minor radiation-induced damage can be compensated for by the reserve progenitor cells of the outer root sheath so that dystrophic anagen responses can be successfully repaired after low doses, and the same anagen phase can be continued. However, this auto-reparative attempt fails at higher doses, and the anagen HFs evolve via a shortened dystrophic catagen reaction and telogen phase into the accelerated establishment of new, largely undamaged HFs that can produce normal hair shafts. At higher radiation doses, permanent HF loss with persistent alopecia may arise due to the irreversible loss of the bulge HFSCs and/or the disruption of the HFSC niche. Following IR exposure, premature senescence in the follicular keratinocytes modulates the secretion of inflammatory mediators and thus the microenvironment in the HF stem cell niches, which has a decisive impact on the regeneration capacity of the bulge HFSCs [37].

### 5.4. Radiation Injury to the Hair Pigmentary System

Graying of the hair is a typical radiation reaction and results from radiation damage to the hair pigmentary system of the HFs. Melanocyte stem cells, located in the bulge regions of both mouse and human HFs, are activated in early anagen, producing melanocytes for reconstructing new HF pigmentary units in the hair bulbs [70]. In the anagen phase, both mouse and human HFs contain mature melanocytes in the hair matrix, whereas amelanotic melanocytes are also observed in the outer root sheaths and hair bulbs [70]. Natural hair graying is a gradual, initially reversible process that is characterized by the defective production and transfer of melanosomes, lighter hair pigmentation and reduced mature melanocytes in the HF pigmentary units [70,71]. Radiation-induced human hair graying likely results initially from damage to the intimate crosstalk between the matrix keratinocytes and melanocytes in the hair bulb, which relies on both epithelial and mesenchymal signals for hair pigmentation [70,72]. Ultimately, the self-renewal of the melanocyte stem cells is impaired by radiation damage, leading to their exhaustion, and thus causing irreversible hair graying [73]. Overall, HFs are an excellent model for studying radiation damage and subsequent organ regeneration through various types of stem cells. The inadequate repair and regeneration of various HF cell populations following IR exposure can lead to HF miniaturization or even HF loss with persistent alopecia.

## 6. Conclusions and Perspectives

Skin epidermis represents the primary physical barrier to the external environment and provides a first line of defense against various physical, chemical and infectious threats. Due to their strategic positioning at the interface between the organism and its environment, epidermal keratinocytes receive signals from the environment and transmit them to the immune cells in the skin. This communication is achieved through the epithelial production of pro-inflammatory mediators, suggesting that keratinocyte-intrinsic signaling pathways play a key role in regulating immune homeostasis during skin inflammatory responses. Recent studies have shown that after IR exposure, radiation-induced senescence of the epidermal keratinocytes and their senescence-associated secretion of inflammatory mediators play a crucial role in triggering radiation dermatitis. As part of the epithelial stress response, the senescence-associated secretion of pro-inflammatory mediators leads to the activation of skin-resident and the recruitment of skin-infiltrating immune cells within the epidermis and dermis, thereby promoting skin inflammation. Accordingly, radiation-induced senescence of the epidermal keratinocytes is a potent mechanism for triggering inflammation, whereby not only the structural but especially the immunoregulatory barrier function of the epidermis is crucial to the development of radiation-induced skin reactions. In addition to acute radiation effects, it must be taken into account that the immune system in the irradiated skin can also be destabilized in the long term, so that either reduced or exaggerated immune reactions can occur after a wide variety of stressful stimuli [74]. However, due to the complexity of skin immune responses, further investigation into the crosstalk that occurs between the epithelial, stromal and immune cell populations will be required to better elucidate the mechanisms that regulate skin immune homeostasis and inflammation.

## Figures and Tables

**Figure 1 ijms-25-03320-f001:**
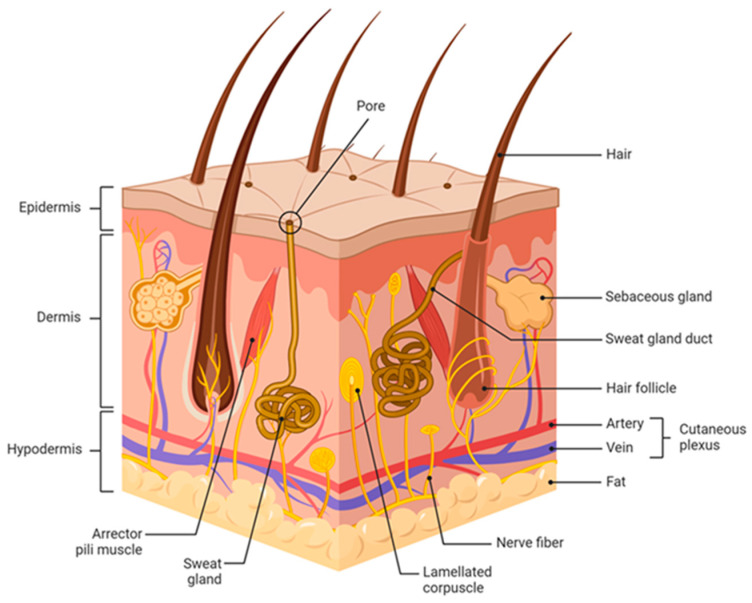
Anatomy of human skin.

**Figure 2 ijms-25-03320-f002:**
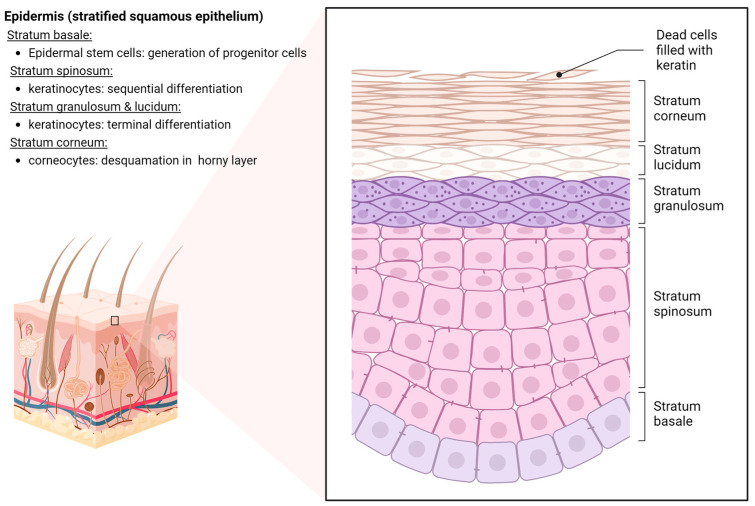
Anatomy of human epidermis.

**Figure 3 ijms-25-03320-f003:**
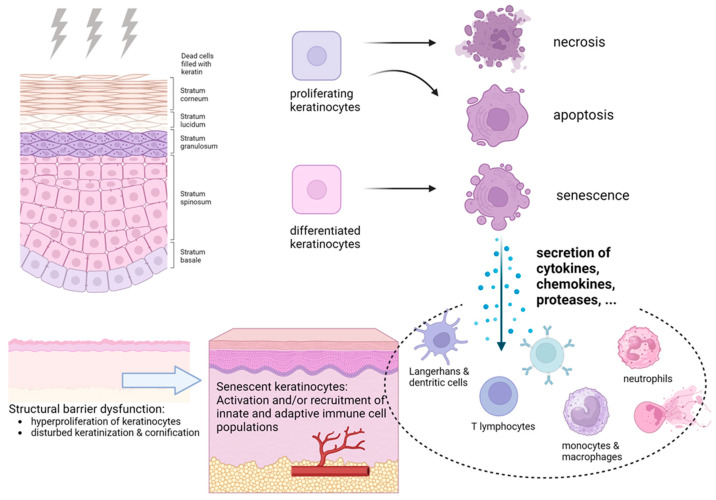
Pathophysiology of radiation-induced skin reactions.

**Figure 4 ijms-25-03320-f004:**
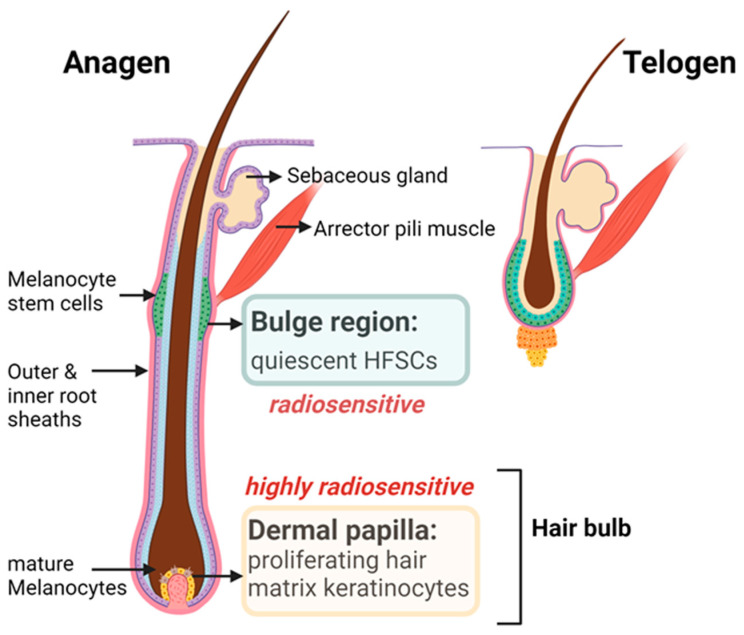
Hair follicle with radiosensitive hair follicle stem cells (HFSCs).

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
