# Peer review of "Radiation Dermatitis: Radiation-Induced Effects on the Structural and Immunological Barrier Function of the Epidermis"

_ijms, 2024, doi:10.3390/ijms25063320_

Round 1

Reviewer 1 Report

Comments and Suggestions for Authors

Overall, the document offers a comprehensive overview of radiation dermatitis, focusing on its pathogenesis and associated factors. It effectively summarizes key aspects of radiation dermatitis and provides valuable insights into its clinical implications. Additionally, the document appropriately emphasizes the importance of further research to understand the complex interplay between epithelial, stromal, and immune cell populations in regulating skin immune homeostasis and inflammation, thus highlighting avenues for future investigation.

The organization of the document is clear, with well-defined section headings that facilitate navigation through different aspects of radiation dermatitis. The content appears to be technically accurate, supported by relevant references and citations to reputable sources, thereby enhancing the credibility of the information presented.

It is commendable that the document acknowledges the complexity of radiation-induced skin damage and emphasizes the need for a deeper understanding of its pathogenesis at the cellular and molecular levels. However, there are some areas where improvements could be made for better readability and clarity.

For instance, certain paragraphs are overly dense and could benefit from breaking down complex concepts into smaller, more digestible chunks. Specifically, the section on "Immune reactions of the skin following IR exposure" could be condensed for better comprehension. 

Additionally, while figures are referenced in the text, sources for these figures are not provided. Including figure sources would enhance the transparency and credibility of the document.

There is one mechanism which is not mentioned in the manuscript. This is skin injury caused by radiotherapy, which can provoke the appearance of new antigens and trigger an autoimmune response. For example, this can occur in conditions such as pemphigoid induced by radiotherapy. Although the association between radiotherapy and bullous pemphigoid has been well documented, the underlying pathophysiology remains uncertain. Multiple hypotheses have been described and partially supported by isolated pieces of evidence. One theory is that  radiation-induced apoptosis of skin epithelial cells causes them to release BP 180 and BP 230 antigens. These endogenous antigens are then recognized by Langerhans cells, which are thought to be resistant to radiation-induced apoptosis, and the production of autoantibodies ensues. Ruocco described a unifying explanation for the occurrence of localized skin diseases secondary to local dysregulated immunity in the context of regional chronic lymphedema, herpes-infected sites, or otherwise damaged areas, which include irradiated skin.

In conclusion, the document serves as a valuable resource for understanding radiation dermatitis and provides a comprehensive overview of its etiology. With minor improvements in language clarity and the inclusion of figure sources, it can further enhance its effectiveness as a manuscript. Additionally, strengthening the conclusion by providing specific recommendations or hypotheses for future studies would improve the overall impact of the document.

Author Response

REVIEWER 1:

Comments and Suggestions for Authors

Overall, the document offers a comprehensive overview of radiation dermatitis, focusing on its pathogenesis and associated factors. It effectively summarizes key aspects of radiation dermatitis and provides valuable insights into its clinical implications. Additionally, the document appropriately emphasizes the importance of further research to understand the complex interplay between epithelial, stromal, and immune cell populations in regulating skin immune homeostasis and inflammation, thus highlighting avenues for future investigation.

The organization of the document is clear, with well-defined section headings that facilitate navigation through different aspects of radiation dermatitis. The content appears to be technically accurate, supported by relevant references and citations to reputable sources, thereby enhancing the credibility of the information presented.

It is commendable that the document acknowledges the complexity of radiation-induced skin damage and emphasizes the need for a deeper understanding of its pathogenesis at the cellular and molecular levels. However, there are some areas where improvements could be made for better readability and clarity.

For instance, certain paragraphs are overly dense and could benefit from breaking down complex concepts into smaller, more digestible chunks. Specifically, the section on "Immune reactions of the skin following IR exposure" could be condensed for better comprehension. 

Response: This section in particular describes the latest findings on the very complex immune reactions after radiation and is therefore the heart of this review. Condensing this section would compromise the comprehensibility and deep insights into the molecular disease processes.

Additionally, while figures are referenced in the text, sources for these figures are not provided. Including figure sources would enhance the transparency and credibility of the document.

L444-445: All figures were created by C.E.R. using the scientific image and illustration software Biorender™.

There is one mechanism which is not mentioned in the manuscript. This is skin injury caused by radiotherapy, which can provoke the appearance of new antigens and trigger an autoimmune response. For example, this can occur in conditions such as pemphigoid induced by radiotherapy. Although the association between radiotherapy and bullous pemphigoid has been well documented, the underlying pathophysiology remains uncertain. Multiple hypotheses have been described and partially supported by isolated pieces of evidence. One theory is that  radiation-induced apoptosis of skin epithelial cells causes them to release BP 180 and BP 230 antigens. These endogenous antigens are then recognized by Langerhans cells, which are thought to be resistant to radiation-induced apoptosis, and the production of autoantibodies ensues. Ruocco described a unifying explanation for the occurrence of localized skin diseases secondary to local dysregulated immunity in the context of regional chronic lymphedema, herpes-infected sites, or otherwise damaged areas, which include irradiated skin.

L431-434:In addition to the acute radiation effects, it must be taken into account that the immune system in the irradiated skin can also be destabilized in the long term, so that either reduced or exaggerated immune reactions can occur after a wide variety of stressful stimuli (Ruocco E. et al. 2014).

In conclusion, the document serves as a valuable resource for understanding radiation dermatitis and provides a comprehensive overview of its etiology. With minor improvements in language clarity and the inclusion of figure sources, it can further enhance its effectiveness as a manuscript. Additionally, strengthening the conclusion by providing specific recommendations or hypotheses for future studies would improve the overall impact of the document.

Reviewer 2 Report

Comments and Suggestions for Authors

This review discusses the intricate pathogenesis of radiation dermatitis, highlighting the limitations of the traditional understanding centered on impaired stem/progenitor cell mitosis. It delves into recent findings indicating that ionizing radiation induces cellular senescence in epidermal keratinocytes, triggering the secretion of pro-inflammatory mediators and subsequent inflammation. The interaction between keratinocytes and immune cells orchestrates the inflammatory response, involving cytokines, chemokines, and the recruitment of neutrophils, exacerbating the inflammatory state. Furthermore, IL-17 expressing γδ-T cells contribute to keratinocyte hyperproliferation, disrupting epidermal barrier function. The conclusion emphasizes the pivotal role of epidermal keratinocytes as mediators between the environment and immune cells, particularly in inflammatory responses. It underscores the significance of radiation-induced senescence in triggering inflammation and the crucial balance between structural and immunoregulatory functions of the epidermis. The paper calls for further investigation into the complex interactions among epithelial, stromal, and immune cell populations to better understand skin immune homeostasis and inflammation regulation in radiation-induced skin reactions.

1. On page 4 (lines 118 – 121), the author states, 'Radiation quality, total dose, dose rate, dose per fraction, as well as the volume and surface area exposed to IR, influence the risk of radiation dermatitis.' However, the information provided appears uncertain. It would be beneficial to expand the paragraph with additional details regarding the quality of radiation and the specific doses that may lead to radiation dermatitis.

2. At the beginning of the paper, the author stated that this review would cover mechanisms of radiation-induced skin damage based on the latest findings. However, specific examples of recent studies were not adequately discussed in this paper. For instance, on lines 122-123, the authors mentioned that 'Different skin areas of the body respond differently to IR, with the most sensitive skin areas being the front of the neck, chest, abdomen, and flexor sides of the extremities.' It would be beneficial to include examples of research data explaining why these areas are more sensitive to IR."

3. In this review, the structure of the skin and the pathogenesis of radiation-induced skin injury are well documented. However, it is considered necessary to add examples of actual research cases for greater clarity

Comments on the Quality of English Language

The English writing is fairly good.

Author Response

This review discusses the intricate pathogenesis of radiation dermatitis, highlighting the limitations of the traditional understanding centered on impaired stem/progenitor cell mitosis. It delves into recent findings indicating that ionizing radiation induces cellular senescence in epidermal keratinocytes, triggering the secretion of pro-inflammatory mediators and subsequent inflammation. The interaction between keratinocytes and immune cells orchestrates the inflammatory response, involving cytokines, chemokines, and the recruitment of neutrophils, exacerbating the inflammatory state. Furthermore, IL-17 expressing γδ-T cells contribute to keratinocyte hyperproliferation, disrupting epidermal barrier function. The conclusion emphasizes the pivotal role of epidermal keratinocytes as mediators between the environment and immune cells, particularly in inflammatory responses. It underscores the significance of radiation-induced senescence in triggering inflammation and the crucial balance between structural and immunoregulatory functions of the epidermis. The paper calls for further investigation into the complex interactions among epithelial, stromal, and immune cell populations to better understand skin immune homeostasis and inflammation regulation in radiation-induced skin reactions. 

  1. On page 4 (lines 118 – 121), the author states, 'Radiation quality, total dose, dose rate, dose per fraction, as well as the volume and surface area exposed to IR, influence the risk of radiation dermatitis.' However, the information provided appears uncertain. It would be beneficial to expand the paragraph with additional details regarding the quality of radiation and the specific doses that may lead to radiation dermatitis.

The exact dose-dependent skin effects are described in the section 3. Clinical features of radiation-induced skin reactions:

 L128-L140: Acute radiation dermatitis usually develops gradually during conventionally fractionated radiotherapy, with the timing of onset varying from days to weeks after the start of radiation treatment. The changes in the skin depend not only on the IR parameters but also on the individual´s skin sensitivity (28), and can include erythema, edema, changes in pigmentation, hair loss and dry or moist desquamation (22). In external beam RT with kilovoltage x-rays, erythema occurs at skin doses ≥6 Gy, dry desquamation at skin doses ≥20 Gy, and moist desquamation at skin doses ≥30 Gy (22). Radiation of higher energies requires higher doses to produce the same level of skin damage, as the maximum dose is received in deeper tissues below the skin. Acute dermatitis typically progresses for 7 to 10 days after RT ends, after which re-epithelialization begins unless a bacterial infection is present. Small areas of skin tolerate higher doses than large regions, as the epidermis can heal by keratinocytes migrating from surrounding healthy skin during the recovery phase (29).

  1. At the beginning of the paper, the author stated that this review would cover mechanisms of radiation-induced skin damage based on the latest findings. However, specific examples of recent studies were not adequately discussed in this paper. For instance, on lines 122-123, the authors mentioned that 'Different skin areas of the body respond differently to IR, with the most sensitive skin areas being the front of the neck, chest, abdomen, and flexor sides of the extremities.' It would be beneficial to include examples of research data explaining why these areas are more sensitive to IR."

This is a general clinical observation in patients, but the exact molecular pathomechanisms have not yet been investigated. However, we were able to show at the cellular and molecular level that for example the foreskin is not only structured differently compared to abdominal skin but also reacts differently to radiation (Hippchen Y. et al. 2022).

  1. In this review, the structure of the skin and the pathogenesis of radiation-induced skin injury are well documented. However, it is considered necessary to add examples of actual research cases for greater clarity

Reviewer 3 Report

Comments and Suggestions for Authors

It is my great pleasure to have an opportunity to review this manuscript. The authors reviewed radiation-induced effects on the structural and immunological barrier function of the epidermis in this manuscript. It is fascinating and well-written. I have some comments and questions that will help you revise your manuscript better.

1.      Radiation dermatitis is classified into acute and chronic radiation dermatitis. In this paper, radiation-induced skin reactions are focused on the acute reactions. The author should describe the chronic reaction in more detail.

2.      I don’t think the chapter “Anatomy and physiology of human skin” is indispensable to this article because there are almost no novel findings in the chapter. It should be written more concisely.

Author Response

It is my great pleasure to have an opportunity to review this manuscript. The authors reviewed radiation-induced effects on the structural and immunological barrier function of the epidermis in this manuscript. It is fascinating and well-written. I have some comments and questions that will help you revise your manuscript better.

  1. Radiation dermatitis is classified into acute and chronic radiation dermatitis. In this paper, radiation-induced skin reactions are focused on the acute reactions. The author should describe the chronic reaction in more detail.

Here we actually focused on the acute radiation reactions of the skin. The long-term consequences of skin irradiation should be described in a separate review article.

  1. I don’t think the chapter “Anatomy and physiology of human skin” is indispensable to this article because there are almost no novel findings in the chapter. It should be written more concisely.

You can only truly understand the complex pathophysiology of radiation dermatitis if you know the exact anatomy and physiology of the skin. Therefore, this chapter should not be removed.